# Application of Elemental Analysis via Energy Dispersive X-ray Fluorescence (ED-XRF) for the Authentication of Maltese Extra Virgin Olive Oil

**Frederick Lia [1,\*], Marion Zammit Mangion [2]**  **and Claude Farrugia [1]**

[1]  Chemistry Department, University of Malta, MSD 2080 Msida, Malta; claude.farrugia@um.edu.mt
[2]  Department of Physiology and Biochemistry, University of Malta, MSD 2080 Msida, Malta; marion.zammit-mangion@um.edu.mt
\*  Correspondence: fredericklia@gmail.com

**Abstract:** Elemental analysis using energy-dispersive X-ray fluorescence on extra virgin olive oils and seed oils revealed the presence of two major concentration related clusters, one containing elements of pedological origin, whilst the other consisted of heavy metals. Seed oils were found to contain a higher concentration of titanium when compared to extra virgin olive oils, whilst extra virgin olive oils derived from the Maltese Islands had a significantly higher concentration of barium and phosphorus on using the Kruskal–Wallis one-way ANOVA ($p$-value < 0.05 for both elements). Application of stepwise linear canonical discriminate analysis proved to be highly superior to PCA, as it was able to distinguish between seed oils from extra virgin olive oils and distinguish between foreign and locally produced extra virgin olive oils.

**Keywords:** extra virgin olive oils; Maltese Islands; elemental analysis; XRF

## 1. Introduction

The overall quality of olive oils is generally defined by both organoleptic properties and oxidation parameters, as defined by the European legislation European Economic Community (EEC) Regulation 2568/1991 [1]. The EEC established a set of rules which define the organization of olive oils. Specifically, Article 35 of the legislation defines the different olive oils according to their free acid content, expressed as oleic acid w/w and organoleptic parameters, whereby extra virgin olive oil (EVOO) may be defined with perfect sensory characteristics when the content of free fatty acids does not exceed 1 g/100 g. In addition, both the European Union Commission No 299/2013 [2] and the International Olive Oil Council [3] have established a knowledge base covering methods and detection limits for the physicochemical parameters of olive oil. This serves to protect the very notion of quality and authenticity against fraudulent activities. The typical methods tend to be focused on the analysis of the fatty acid constituents through profiling and have generally overlooked the analysis of mineral constituent in EVOOs. Although such methods tend to be particularly useful in the detection of adulterants in EVOOs, these methods tend to be rather time-consuming. Furthermore, these methodologies are not sufficiently effective in distinguishing geographical origins.

It is well-known that the inorganic content of olive oils has an important role in terms of food safety and shelf-life. Trace heavy metals in vegetable oils are known to influence the rate of oil oxidation, where oxidation leads to the development of unfavorable odors and taste, causing the deterioration of olive oils. The major factors that affect the rate of oxidation are the degree of unsaturation, the amount of oxygen, the temperature, the amount of light and the presence of metals (mainly transition metals, such as Fe and Cu) [4,5]. Benedet and Shibamoto showed that trace amounts of Fe, Cu, Cr, Pb and

Cd contribute oxidative effects to lipid peroxidation [6]. The presence of these trace metals enhanced the rate of oxidation of edible oils by increasing the generation of free radicals from fatty acids and hydroperoxides.

The presence of metals in vegetable oils can be attributed to two major uptake pathways which may be endogenous or exogenous. Endogenous pathways are connected with the plant metabolism whereby inorganic constituents are incorporated into the oil by the natural uptake and preconcentration of the element by the plant. Exogenous pathways are attributed to contamination during the production and the collection of olives and seeds during the oil extraction and treatment processes, (by processing actions such as bleaching, hardening, refining and deodorization) [7]. Elements may also be introduced by systems and materials of packaging and storage, for example, from foreign bodies during harvesting or wear metals in the press [8,9]. The determination and the analysis of trace metals offers a challenge mainly because of the hard organic content of the oil matrix. Analytical techniques have been employed for the determination of metals in oils and rely on both emission and absorption spectrophotometric techniques, as well as electroanalytical techniques. [8,10–12]. Traditional methods for elemental analysis include inductively coupled plasma optical emission spectroscopy (ICP-OES), atomic absorption spectrometry (FAAS/GFAAS) and ion chromatography (IC). The application of X-ray fluorescence (XRF) has been shown to be very important for the determination of trace, minor and major elements in a large variety of matrices [13]. XRF has a number of advantages over other techniques: It uses very small amounts of the test material, requires minimal sample preparation and offers a nondestructive quantification method. On the other hand, the determination of organic content by XRF is still considered a difficult task since the X-ray cross-sections for light elements are very low [14]. The main drawback of X-ray analysis is that it cannot be used for the determination of elements lighter than sodium. This is attributed to the low penetrating power of the secondary radiation from lighter elements. The presence of these elements tends to be highly elusive, as X-rays emitted from these elements are usually attenuated before they reach the detector and are potentially masked by background radiation from the X-ray emission source [14].

In this study, XRF was performed by using ED-XRF (energy dispersive XRF) devices, which can provide a semiquantitative analysis with minimal sample preparation. The major drawback of such a method, when compared to other methods, such as inductively coupled plasma mass spectrometry (ICP-MS) and atomic absorption spectroscopy (AAS), is its low accuracy in identifying oils derived from the same geographical origin and soils. However, since this study focuses on the development of rapid and cheap methods for the discrimination of Maltese EVOOs from foreign EVOOs, as well as seed oils, such a method was deemed appropriate for the aim of the study. The aim of this study was to establish whether XRF fluorescence could be used for the authentication of Maltese EVOOs. The hypothesis behind the application of elemental analysis of olives and its direct reflection of the corresponding geographical origin is based on the different mineral content of soils which are later incorporated into the plant tissues and ultimately in the oils.

## 2. Materials and Methods

### 2.1. Sample Preparation

For this study, a total of 54 extra virgin olive oil samples were collected over 4 harvest seasons, from 2013–2016. Twenty-two samples were collected from the Maltese Islands, and 32 were collected from other neighboring Mediterranean countries. The samples were all taken from different oil producers, in order to cover a representative sample of the Maltese Islands in terms of pedological and microclimatic conditions and also of manufacturing techniques and different presses employed. Foreign olive oils obtained were bought with a protected designation of origin, in order to ensure traceability of the product. A number of different refined seed oils, namely sunflower, rapeseed, peanut and soybean oil samples, were also purchased from local stores (3 samples of each). Olive oils were stored at 4 °C, in the absence of light, prior analysis. The samples were preheated to 35 °C in a water

bath, for an hour, and mixed to ensure homogeneity; 25 g of each sample was transferred to XRF sample cups with a 4 μm Prolene® film as a base.

## 2.2. XRF Measurements and Data Acquisition

Each sample was analyzed, using a Bruker S2 Ranger (Bruker Corporation, and Madison, WI, USA), at tube voltages of 10, 20, 40 and 50 kV, for 100 seconds, at each voltage under a helium atmosphere; in the case of the 4 0kV voltage, an aluminum filter was used, while at the 50 kV voltage, a copper filter was used. The concentrations were then determined by using a fitting method supplied by Bruker, where the sample matrix was set as fructose. For each scan, it was determined that the fitting parameter $R/R_0$ was less than 10, indicating good fit. Each sample was run in triplicate, and the concentrations obtained were averaged. Correlation analysis and Kruskal–Wallis ANOVA were carried out by using IBM SPSS statistics 24, whilst PCA and SLC-DA was carried out by using the JMP 10 (SAS). Elements wherein the concentration was consistently low for all samples (0–3 ppm) were excluded from this study due to instrumental limitations.

## 2.3. Multivariate Statistical Analysis

### 2.3.1. Principal Components Analysis (PCA)

PCA is an unsupervised multivariate variable reduction technique that is used to elucidate the relationships between samples which are described by a large number of variables. In PCA, the original set of multidimensional data is orthogonally transformed into a set of linearly uncorrelated variables referred to as principal components (PCs), in such a way as to maximize the variation. Although it is not the main aim of PCA, it also enables the detection of sample patterns and possible clustering of observation and to detect outliers. This is done through the analysis of the sample scores, which give an indication of the sample similarities or differences along a component. Similar scores for different samples show that they are similar along a PC, while different scores mean that the samples are different with regard to that PC. Consequently, scores from two different principal components are usually plotted in bi-plots, in order to give a visual representation of the variation of the data in a score plot. On the other hand, loadings are used to describe the influence each individual variable has on a principal component.

### 2.3.2. Stepwise Linear Canonical Discriminant Analysis (SLC-DA)

SLC-DA is a linear discriminant method that is used for classification and data reduction. In this study, discriminant analysis, using a linear method, was applied. This works by calculating distances from group means as the Mahalanobis distance, whereby a common covariance matrix is used for all groups. A stepwise analysis allows for manual selection of variables used to build the linear model up to a maximum number of entries (n−1), where n is the number of samples in the sample set. This enables the selection of variables with large F-ratios and small *p*-values, in order to build a model on the most discriminant variables. With the addition of a new variable, the values for F and *p* will change for both the included and excluded variables, with the termination step being when the F-ratio and *p*-value approach a value of 0 and 1, respectively. Canonical discriminant analysis generates components referred to as canonical discriminant functions that are described by scores and loadings. Similar to PCA, the first two canonical functions provide the maximum separation amongst different groups. In order to prevent the model form overfitting, the samples where split into two groups which correspond to the training and validation sets. A sample splitting scheme was adopted in order to cover such variation in the two sets. The oil samples were grouped in an ascending way, so that the first 22 samples would represent Maltese EVOO's, whilst the rest correspond to non-Maltese EVOO's and seed oils. The dataset was split in such a way that every $s^{th}$ ($s = 5$) sample was omitted from the dataset. This validation method excluded 20% of the observation, so that the 20% would be retained as the validation set. The remaining 80% of the observation was used to build the training set.

## 3. Results and Discussion

### 3.1. Elemental Correlation Analysis

A correlation matrix analysis was carried out on the % abundance of all the elemental constituents identified through ED-XRF. Given that the data observed were multivariate non-normally distributed, the correlation analysis was carried out by using Spearman's Rho rather than Pearson's. The analysis shown in Figure 1 revealed the existence of two major elemental positively correlated clusters found in oils. The first cluster consisted of Si, P, S, Al, Mg and Ca, which are most likely to be derived from the pedological and biological origin. The second correlation cluster consisted of heavy-metal elementals, including Fe, Cr, Zn, Cu, Ni and Co. The significant positive correlation between each element suggests that they tend to originate from a source, in which the concentration of these elements has a fixed composition. However, one must exclude that the plant is simultaneously bioaccumulating these elements at different rates, irrespective of the individual concentration in the soil. Many species of plants have been shown to be capable of taking up heavy metals from soils that are essential for plant growth (Fe, Mn, Zn, Cu, Mg, Mo and Ni), and other metals with unknown biological function (Cd, Cr, Pb, Co, Ag, Se, Hg) have also been found to accumulate [15]. Only very few bioaccumulation experiments have been performed using *Olea europaea*, mainly because research is focused on the use of non-edible crops for the extraction of heavy metals from contaminated soils. A study conducted by Aghabarati, Hosseini and Maralian [16] demonstrated that olive trees irrigated with municipal effluent increased the levels of Zn, Pb, Cr and Ni in both soil and plant, but were below the permissible limits. Similar results were observed by Llorent-Martínez et al. [17], whereby the positive correlation between the concentration of Cr and Fe was attributed to the production procedure, which can involve stainless-steel mechanisms containing the two elements. However, in the same study, no correlation between the concentrations of Cr and Fe was found with other heavy metal elements present in oils, such as Mn and Cu.

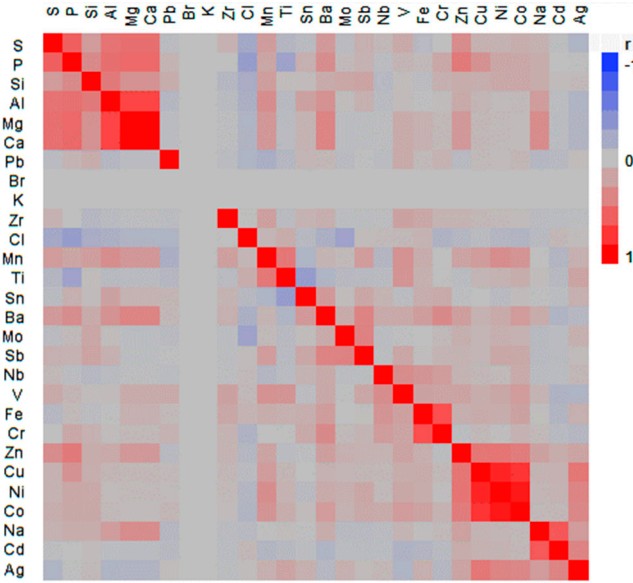

**Figure 1.** Clustered correlation analysis of elemental analysis present in olive oils and other seed oils. Common earth elements; sulfur, phosphorous, silicon, aluminum, magnesium and calcium show a positive correlation with each other. Similarly, heavy metals, including iron, copper, zinc, chromium, nickel and cobalt, tend to show a positive correlation with each other.

### 3.2. Discriminating EVOOs from Refined Seed Oils on the Basis of Elemental Composition

Trace-element analysis also plays an important role as a basis for oil adulteration detection and oil quality control [17,18]. Kruskal–Wallis one-way ANOVA revealed a significant difference in the

percentage abundance of the Ba, Cl, Ti, Co, Sn and P across olive oils of different geographical origin and other refined oils used in this study. The Mann–Whitney pairwise comparison also showed that % abundance of titanium was significantly ($p$-value < 0.01) higher in refined oils when compared to all the EVOOs. This may be attributed to the extensive treatment and processing to which refined oils are subjected, including bleaching, refining and deodorization. These treatment process could increase the risk of heavy-metal incorporation in the oil matrix. In this study, no significant difference was observed in the percentage abundance of other heavy metals that are typically used in the seed oils refining processes, such as copper and nickel, which are typically used as hydrogenation catalysts, [11] or chromium and iron, which are potential contaminants of the oil deriving from the processing equipment [19].

### 3.3. Application of ED-XRF for Maltese EVOO Authentication

The determination of the geographical origin of extra virgin olive oils in order to assess traceability can be controlled by chemical species that are linked to the production area. The hypothesis is that the transfer of elements from the soil to the olive tree, and later to the oil, is subjected to minor variations in soil and geology characteristics, and thus the elemental composition of EVOOs is a direct reflection of its geographical origin. The minor variations in the elemental constituents could provide information that can be used for geographical traceability [12,20]. Figure 2 revealed that olive oils derived from both indigenous and other local cultivars had a significantly higher percentage abundance of barium and phosphorus. This observation suggests that the concentration of these elements may be used as a typical marker for the determination of origin. The presence of a significantly higher barium content in olive oils of Maltese origin can probably be attributed to pedological effects. Barium is not very mobile in most soil systems, due to the formation of water-insoluble salts and an inability of the barium ion to form soluble complexes with fulvic and humic acids. The presence of organic material, calcium carbonate, chlorides and soil pH affect the availability of barium. In soils with a high organic-matter content, barium mobility is limited [21,22]. High $CaCO_3$ and sulphate salts content also limits mobility, by precipitation of barium carbonate and sulphate [23,24]. However, in the presence of chloride ions, barium is more mobile and is more likely to be incorporated into the plants due to the high solubility of barium chloride as compared to other chemical forms of barium [21,23]. The availability of barium in Maltese soils is thus dependent on two opposing effects, the moderately alkaline (pH between 7.3 and 8.5) and the high calcareous content, which reduce the mobility of barium, whilst the low organic content [25] and relatively high concentration of chloride ions derived from marine origin increase the mobility of barium. Although the presence of calcium carbonate reduces the availability of barium to the plants, its presence fixes the barium content in Maltese soils, preventing it from leaching out of the solution.

In the study carried out by Camilleri and Vella in 2010 [26], it was shown that the aerial barium concentration in the Maltese Islands significantly increases during July and August. This seasonally dependent emission was attributed to the burning of fireworks during the summer period, which also coincides with the olive fruit maturation stage. Camilleri and Vella, in 2010 [26], also showed that the content of Ba persists even during the latter part of the summer. Barium from fireworks is expected to be particularly bioavailable, since it consists of water-soluble species, such as $BaCl_2$, $BaO$, $Ba(OH)_2$ and residual $Ba(NO_3)_2$ [27]. The significantly higher concentration of phosphorus in olive oils of Maltese origin may be related to a number of different effects, mainly the extensive use of fertilizers 26 kg/ha [28] and the high pH of the soil. Similar to barium, the presence of calcium and the moderately alkaline pH across all Maltese soils prevents phosphorus compounds from leaching. Unlike barium, in the case of phosphorus, the incorporation of the plant material is an active process requiring an energy-driven transport mechanism to move phosphorus through the membranes into the plant root cells via phosphate transporters. Thus, although the phosphorus might be chemically fixed, the plant might still be able to strip phosphorous compounds from the surrounding soil, in response to the different pedoclimatic conditions that the plants are subjected to.

Figures 2 and 3 show that the application of multivariate statistical analysis showed the spatial distribution of both olive oils derived from different origins and that an ability to discriminate between extra virgin olive oils and other refined seed oils is possible on the basis of elemental composition analysis. Application of PCA with the aim of exploring the variability in the elemental composition of EVOOs and seed oils was carried out. We applied the PCA to the correlation matrix, which is equivalent to the use of the variance–covariance matrix of standardized variables. From the results obtained, it was shown that the first two principal components only explained 28.17% of the total variance, with a considerable loss of more than 82% of information. The very low explained total variation is an indication that only a relatively small number of variables is actually contributing to the observed variation of data. Therefore, from the 28 elements which were quantified using ED-XRF, only very few of these elements are actually providing information about the variation in the samples. However, not all the elements might be actually responsible for the observed variation from the score plot obtained, as illustrated in Figure 2. A marked grouping in the seed oils and EVOOs easily along the second principal component was observed, whilst not wholly distinct EVOOs of Maltese and non-Maltese origin were also grouped separately along the first principal component. On further analysis, the principal component scores were observed that heavy metal content, namely Co, Ni, Cu and Zn, had the highest Eigenvectors in the first principal component, indicating that the % abundance of these elements in oils offers the largest explanation of covariance. Similarly, Ba and Ag offered the highest % of covariance explained in the second principal component scores.

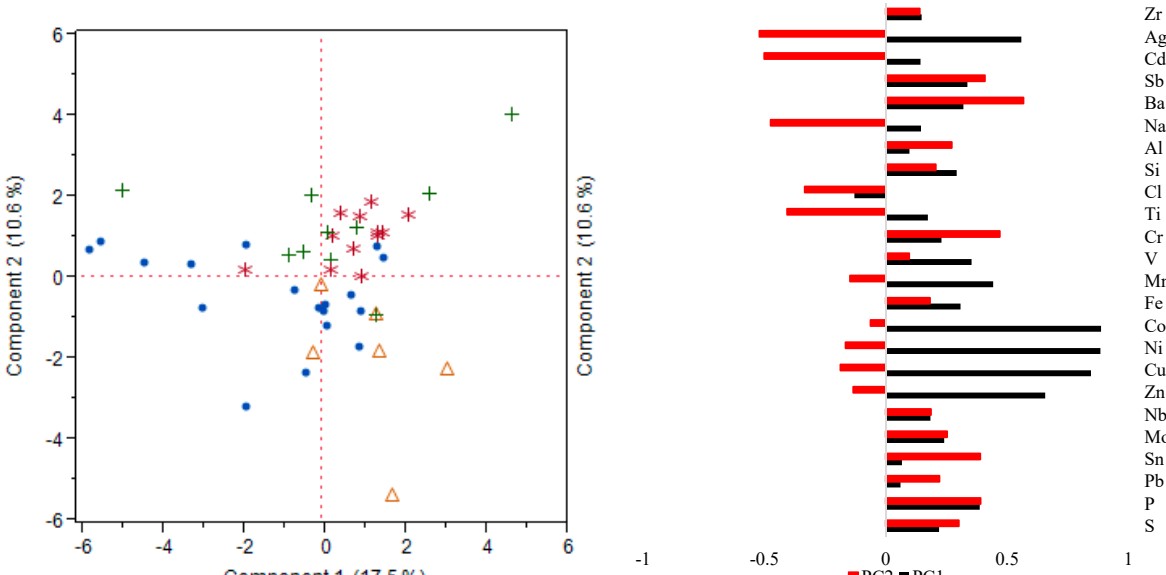

**Figure 2.** Principal component analysis of the elemental analysis of oils derived from different sources and olive oils of different geographical origin. Red stars (*) represent the olive oils derived from indigenous cultivars; green crosses (+) represent locally derived foreign olive oils; blue dots (●) represent foreign olive oils; and orange triangles (Δ) represent oils derived from other seed oils. (Right) Eigenvector loading scores for each element for the first two principal components.

In comparison, Figure 3 shows the results obtained by using SL-CDA, where the model obtained was based on excluding the % abundance K, Ca, Mg and Br, whilst retaining all other elements. The model was able to explain 90.59% of the observed variance over two canonical functions, resulting in an overall 2.17% misclassified observations under full cross-validation. The model was able to fully discriminate between olive oils and refined oils, as well as olive oils of different geographical origins. Figure 3 also illustrates the standardized discriminant function coefficients; these indicate the relative importance of the independent variables in predicting the dependent variable. It was shown that the major elements which contribute to the discrimination of Maltese EVOOs from seed oils

and foreign EVOOS were Ba, Si, Ag and Cl, which contributed mainly to the first canonical function, and Cu, Zn, Ti and Ni, which contributed to the second canonical function. On further inspection of the Wilk's lambda obtained (0.0093, *p*-value < 0.001) a small significant value was obtained, indicating that the function obtained has a good discriminatory power and is able to separate cases into groups. The significant *p*-value obtained for Wilk's lambda indicates that the discriminant function does better than chance at separating the groups.

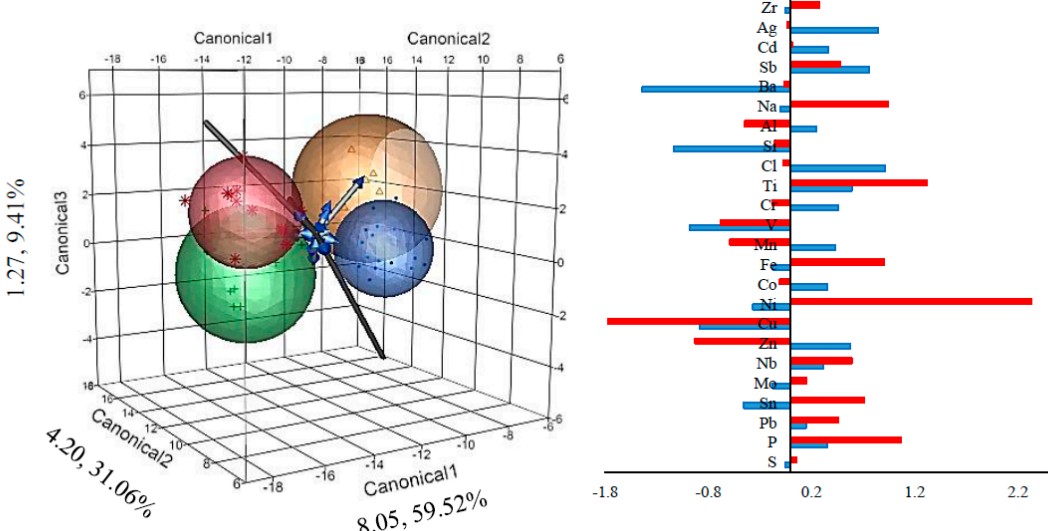

**Figure 3.** (Left) 3D canonical score plot obtained for the elemental analysis using SL-CDA. Values next to the canonical function represent the eigenvalue and the corresponding % of variation explained. Markers represent the same data presented in the previous figure. (Right) Canonical scores for each element, blue bars correspond to the canonical scores obtained for the first canonical function, whilst the red bars represent the scores obtained for the second canonical function.

### 3.4. Discussion

In this study, a multivariate discriminant analysis was performed in order to develop a discriminating function which can help predict olive oil origin based on the % abundance of elements. Similar studies conducted by Cabrera-Vique et al. (2012) [29] have shown that elemental analysis of extra virgin olive oil from PDO's of Granada and Jaén had a significantly different concentration in the Ni, Mn, Cu, Cr and Fe content. Furthermore, in this study, the values of the five trace elements had the highest potential as predictor variables in the model built using multiple linear analysis. The importance of these heavy-metal elements in the discrimination of olive oils of different geographical origin is concordant with the results obtained by other authors Benincasa et al. (2007) [12] and Zeiner et al. (2010) [20]. Zeiner et al. (2010) reported that the determination of the trace-metal pattern of olive oils by atomic spectrometric methods showed that the concentrations of Cu, Mn and Ni vary significantly by geographical origin of olive oils obtained from Croatia and can be used for geographical characterization of the oils. Similar to the presented study, the concentrations of Fe, Mg, Na, and Ca in the samples showed no significant differences according to the geographical origin of the oils. Zeiner et al. (2010) [20] further suggested that higher metal levels in olive oils may be an indication of improper production of extra virgin olive oils and therefore jeopardize the results for a certain geographical region. Benincasa et al. (2007) [12] used linear discriminant analysis (LDA) in order to distinguish between olive oils derived from three Italian geographical locations for two different cultivars, Coratina and Carolea. Similar to our study, from the available data, it appears that the inter-cultivar variation and olive oils derived from indigenous and imported locally grown cultivars cannot be distinguished by elemental composition, since they are grown under the same

geographical conditions. This suggests that the terroir has a large effect on the elemental composition of the olive oil rather than the cultivar itself.

## 4. Conclusions

In this study, semiquantitative techniques were used, whereby the % abundance of each element in the sample was determined in EVOOs. The use of ED-XRF for the determination of the elemental composition of EVOOs proved to be an effective alternative to conventional absorption spectrometry. The additional advantages of XRF lay in the simplicity of the spectra obtained, minimal sample preparation and the nondestructive nature of the analysis. In the study, we were able to show a significant positive correlation between elements of pedological origin and heavy metal elements, which can be both of botanical origin or derived from anthropological sources. The method may be ideal as a first rapid test to distinguish EVOOs grown in a given terroir, although it is less useful where inter-cultivar variation and olive oils derived from indigenous and imported locally grown cultivars, since they are grown under the same geographical conditions. This study suggests that, based on the elemental composition, the concentration of Ba, P, Co, Ni, Cu and Zn can be employed for the discrimination of olive oils of Maltese origin.

**Author Contributions:** F.L., data acquisition, research paper conceptualization, methodology, software, validation, formal analysis, investigation, data curation, writing—original draft preparation, writing—review and editing, and funding acquisition; M.Z.M., conceptualization, writing—original draft preparation, writing—review and editing, and supervision; C.F., conceptualization, software, writing—original draft preparation, formal analysis supervision and project administration. All authors have read and agreed to the published version of the manuscript.

**Funding:** This research was funded by e Malta Government Scholarships Post-Graduate Scheme for 2014 (MGSS-PG 2014).

**Acknowledgments:** The research work disclosed in this publication is funded by the Malta Government Scholarship Scheme 2014. Special thanks go to Joseph Grech for his support and maintenance of the instrumentation.

**Conflicts of Interest:** The authors declare no conflict of interest.

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
