# Peer review of "Application of Elemental Analysis via Energy Dispersive X-ray Fluorescence (ED-XRF) for the Authentication of Maltese Extra Virgin Olive Oil"

_agriculture, doi:10.3390/agriculture10030071_

Round 1

Reviewer 1 Report

In my opinion, the manuscript might be improved by delivering short descriptions of the techniques used.

I guess, the authors should remove all not common-use acronyms from the abstract.

Author Response

Comments and Suggestions for Authors

In my opinion, the manuscript might be improved by delivering short descriptions of the techniques used.

We have included some more details regarding the use of statistics in the methods section, however all the methods have been described in detail such that they can easily be replicated by other scientists

I guess, the authors should remove all not common-use acronyms from the abstract.

Corrected

Reviewer 2 Report

Overall this is a well written manuscript with an interesting application of XRF for fingerprinting oil origins.  

In terms of the XRF analysis, the lack of validation is a concern as in my experience the matrix will have a significant effect on the accuracy of the result.  As the absolute value of the elements is not a key to this research this should not be a major issue.  However, the conversion to % abundance is a concern as without a validation it is not possible to determine the accuracy and or error in the calibration for each of the elements and it is not possible to determine if any error is consistent across all elements and concentration ranges.    

Furthermore a validation of the SLC-DA model is essential to ensure the validity of this application.  The results do show a suitable differentiation between the geographical locations and oil types when applying the SL-CDA model. Whilst I am not an expert in this area I have found in my experience with some complex models it is possible to achieve a model that appears to be suitably robust, however when applied to a validation set this does not provide suitable prediction accuracy.  Consequently I would suggest a validation of this prediction model is required to improve this manuscript.

Author Response

We perfectly agree with your comment whilst we are perfectly aware of such problem please note that one of the objectives of the study was to develop fast and effective methods for discrimination hence that is why we applied the use of XRF rather than ICP-MS. The latter would provide a more absolute quantitative data. The comparison between the two techniques would be carried out as a separate study in order to determine the errors associated with XRF using ICP-MS as the primary technique through which XRF is validated. However this was far out of the scope of the study.   

We are very sorry for this misconception it was our fault since we did not specify the results reported within the manuscript are those obtained after external validation. We have updated the SL-CDA section regarding the validation procedure which was carried out. We are perfectly aware of model overfitting and how to avoid it. Thanks for bring this to our attention.

Reviewer 3 Report

Manuscript Number: Agriculture

Title: Application of elemental analysis via X-ray fluorescence (XRF) for the authentification of Maltese extra virgin olive oil

The paper is very interesting and it is very important for publication. However, major corrections must be taking into account, as follow:

1.- For study to have a certain scientific rigor, it would be necessary to know the geographical area of origin of the each sample, since this could be correlated with the climate and soil.

2.- Line 52: there are words together: ([7].Elements).

3.- Line 334-335:  there are united references ([9] and [10]).

4.- Line 339: Why stress ¨Microchemical Journal¨.

5.- Line 343-344: Missing magazine number.

6.- There are smaller errata in the references, you should follow the rules of the magazine.

Best wishes

Author Response

1.- For study to have a certain scientific rigor, it would be necessary to know the geographical area of origin of the each sample, since this could be correlated with the climate and soil.

Whilst it is true and would be a very good study to correlate the mineral content of the soil to the mineral content of the fruit and ultimately the oil, the aim of the study was to provide fast and sensitive methods for the authentification of the Maltese olive oils through the use of ED-XRF. Whilst the geographical origin of each oil was identified the  exact pin point of each olive sample within each oil especially the foreign once could not be obtained. In the case of the Maltese islands each olive oil was handpicked from the producer however the problem in the Maltese islands is that, due to the small fields and low yield the olive press tends to combine olives from different farmers into one single press and then divided the oil content  by the initial weight of the olives. Due to these reasons obtaining monocultivar olive oils derived from one single locality proved to be very difficult as it is hampered by the size of the Maltese islands and the constant treat of agricultural land destruction. As is expected this would increase the number of variables within the system, whilst for conventional statistical models this would be devastating. The use of multivariate statistical methods not only avoids the problems but also provides a solution for such variability as seen for the results obtained.

2.- Line 52: there are words together: ([7].Elements).

Corrected

3.- Line 334-335:  there are united references ([9] and [10]).

Corrected

4.- Line 339: Why stress ¨Microchemical Journal¨.

Corrected

5.- Line 343-344: Missing magazine number.

Corrected

6.- There are smaller errata in the references, you should follow the rules of the magazine.

Corrected

Round 2

Reviewer 2 Report

I appreciate the point of this paper is for a rapid analysis method, and indeed XRF is perfect for this purpose.  

However, without first validating the XRF method it is not accurate to report the elemental concentrations/ % abundance without first confirming the XRF method accuracy for these oils. 

One suggestion may be to use the EDXRF intensity rather than reporting a concentration for the elements as the SL-CDA modelling is not dependant on quantitative data rather the elemental fingerprint.

The changes in text have suitable explained any concerns regarding the SL-CDA modelling.

Author Response

That would have been the best way and we tried and even contacted the supplier but it was not possible to extract the raw data from the spectrum and we could only report elemental abundance in terms of % rather than intensity as it was primarily intended. I understand and your point, it is very valid and I can perfectly agree with you however, considering that the model fitting criteria onto which the % of elemental abundance was very high and that replicates for each oil sample where taken this further put confidence in the results obtained. Thanks again for your comment.

Reviewer 3 Report

Manuscript Number: CLAY-D-19-00107

Title: Application of elemental analysis via X-ray fluorescence (XRF) for the authentification of Maltese extra virgin olive oil

The paper is very interesting and it is very important for publication. Most of the corrections have been made and the first question have been justified. Therefore it is considered appropriate to publish in its latest version.

Best wishes

Author Response

Thanks alot no corrections required